# Morphological Medial Gastrocnemius Muscle Growth in Ambulant Children with Spastic Cerebral Palsy: A Prospective Longitudinal Study

**DOI:** 10.3390/jcm12041564

**Published:** 2023-02-16

**Authors:** Nathalie De Beukelaer, Ines Vandekerckhove, Ester Huyghe, Geert Molenberghs, Nicky Peeters, Britta Hanssen, Els Ortibus, Anja Van Campenhout, Kaat Desloovere

**Affiliations:** 1Department of Rehabilitation Sciences, KU Leuven, 3000 Leuven, Belgium; 2Interuniversity Institute for Biostatistics and Statistical Bioinformatics (I-BIOSTAT), KU Leuven, 3000 Leuven, Belgium; 3Interuniversity Institute for Biostatistics and Statistical Bioinformatics (I-BIOSTAT), Data Science Institute, Hasselt University, 3590 Diepenbeek, Belgium; 4Department of Rehabilitation Sciences, Ghent University, 9000 Gent, Belgium; 5Department of Development and Regeneration, KU Leuven, 3000 Leuven, Belgium; 6Department of Orthopedics, University Hospitals Leuven, 3000 Leuven, Belgium; 7Clinical Motion Analysis Laboratory, University Hospitals Leuven, 3000 Leuven, Belgium

**Keywords:** cerebral palsy, ultrasound, piecewise model, muscle volume, cross-sectional area, GMFCS

## Abstract

Only cross-sectional studies have demonstrated muscle deficits in children with spastic cerebral palsy (SCP). The impact of gross motor functional limitations on altered muscle growth remains unclear. This prospective longitudinal study modelled morphological muscle growth in 87 children with SCP (age range 6 months to 11 years, Gross Motor Function Classification System [GMFCS] level I/II/III = 47/22/18). Ultrasound assessments were performed during 2-year follow-up and repeated for a minimal interval of 6 months. Three-dimensional freehand ultrasound was applied to assess medial gastrocnemius muscle volume (MV), mid-belly cross-sectional area (CSA) and muscle belly length (ML). Non-linear mixed models compared trajectories of (normalized) muscle growth between GMFCS-I and GMFCS-II&III. MV and CSA growth trajectories showed a piecewise model with two breakpoints, with the highest growth before 2 years and negative growth rates after 6–9 years. Before 2 years, children with GMFCS-II&III already showed lower growth rates compared to GMFCS-I. From 2 to 9 years, the growth rates did not differ between GMFCS levels. After 9 years, a more pronounced reduction in normalized CSA was observed in GMFCS-II&III. Different trajectories in ML growth were shown between the GMFCS level subgroups. These longitudinal trajectories highlight monitoring of SCP muscle pathology from early ages and related to motor mobility. Treatment planning and goals should stimulate muscle growth.

## 1. Introduction

Skeletal muscle changes are frequently manifested in children with spastic cerebral palsy (SCP) which interact with the neuromotor functioning throughout development [1,2,3]. However, the underlying mechanisms of this potential bilateral causations or associated presence remain unknown. Additionally, neuromotor impairments, such as spasticity and weakness, manifest from young ages, and result in heterogenous clinical presentations of children with SCP [1,4,5,6,7]. To classify the children based on functional abilities in daily life, the Gross Motor Function Classification System Expanded and Revised (GMFCS-E&R) is frequently used [8]. The 5 GMFCS levels are defined using standardized descriptions for which the lower levels indicate higher motor abilities. To reduce and manage the impact of neuromotor impairments on the daily motor functioning, children with SCP are frequently treated at the muscular level (e.g., botulinum neurotoxin injection or casting) from young ages [9]. 

In typical development, morphological muscle growth is triggered by body size increases and responds to the demands through protein synthesis [10,11,12]. However, for children with SCP, previous research highlighted lower increases in medial gastrocnemius (MG) muscle volume (MV) with increasing age, which was unrelated to the rate of skeletal growth and already presented at the age of 15 months [13]. Taking body length and mass into account, MVs were found to reduce by 3% over a 6-month follow-up period in 2–5 years old children with SCP [14]. Over the age ranges of 2 to 12 years, cross-sectional investigations identified normalized medial gastrocnemius MV deficits of 22–57% and shorter normalized muscle belly lengths (MLs) with deficits of 11–19% compared to age-matched typically developing (TD) children [15,16,17,18,19]. Since the muscle is a very plastic tissue, the observed muscular changes in SCP could reflect adaptations to the functional demands including altered muscle tone and limited motor functioning [12]. Indeed, previous cross-sectional studies demonstrated increasing muscle pathology, such as deficits in MV, anatomical cross-sectional area (CSA) and ML, in relation to higher GMFCS levels [8,13,18,20,21]. Yet, research to date has not determined the impact of the severity in motor impairments on the trajectory of altered muscle growth. Overall, these previous cross-sectional observations highlighted significant alterations in the SCP muscle from a young age, which were suggested to be multifactorial in origin, for example due to impaired neural activation, inflammation and altered or non-use of the muscle [3,12,22,23,24]. 

Up to now, cross-sectional investigations have already suggested altered associations between muscle morphology and age in SCP versus TD children [13,16]. These previous studies assumed a linear trajectory of morphological muscle growth with age. However, much uncertainty remains in the variability of the trajectories for altered muscle growth over the time-span from infancy to school ages and between different levels of severity in children with SCP [13,16,25]. Further, MV is frequently reported apart from the estimates of muscle growth in the cross-sectional dimension (e.g., CSA) and longitudinal dimension (e.g., ML). To delineate the muscle growth trajectories, repeated assessments over time per individual are required. These longitudinal datasets generate child-specific profiles per morphological measure. A better understanding of this time course may shed light on opportunities for the planning, development and promotion of muscle growth strategies, whether or not combined with the conventional therapeutic strategies. 

The current prospective longitudinal study investigated morphological MG muscle growth over a wide age span, including infants (6 months–2 years), preschool (2–5 years) and school-aged SCP-children (6–11 years). The first objective was to model the muscle growth trajectory with respect to age by using repeated muscle assessments. Growth in the MG muscle morphology was expressed for different absolute and normalized outcomes. Second, we aimed to compare the trajectories in muscle growth between the GMFCS levels I and II–III. We hypothesized gradually increased morphological muscle outcomes (i.e., muscle volume, mid-belly cross-sectional area and belly length) with increasing age, with lower muscle growth rates in children of the higher GMFCS levels.

## 2. Materials and Methods

### 2.1. Study Design and Participants

This prospective, observational cohort study was designed with a prospective longitudinal research protocol of repeated assessments and was conducted between April 2018 and March 2022. Over a follow-up period of 2 years, predefined intervals of minimal 6 months up to 2 years were set between the repeated assessments. These intervals further depended on the timing of the regular clinical follow-up appointment during which the assessments were performed. As a result, the observational data was included with 2 to 5 assessments per child. Via the Clinical Motion Analysis Laboratory (CMAL-Pellenberg) and Cerebral Palsy Reference Center of the University Hospitals Leuven, children with a diagnosis of SCP confirmed by a pediatric neurologist, were recruited. SCP-children aged between 6 months and 9 years at baseline, with bi- or unilateral involvement and with GMFCS-E&R levels I to III, defined by the criteria per age-band [26], were included. Additionally, the following exclusion criteria were applied at baseline and during follow-up: (a) Botulinum neurotoxin type A (BoNT-A) treatment 10 months prior to the assessment (b) serial casting 3 months prior to assessment and (c) history of orthopedic and/or neurosurgery. Considering potential change in GMFCS level over time, which could be expected by consequence of motor development, especially at the youngest ages, the GMFCS level at the last assessment was used to define subgroups. These subgroups, i.e., GMFCS level I vs. level II & III, were selected to maximize the number of observations per group and to reduce heterogeneity [7,27]. For the latter, it has been described that children with GMFCS level I are more stable in motor function in comparison to children with levels II and III, and this stability is found to be already established by the age of four years [7,27]. To illustrate muscle deficits in the children with SCP, a retrospective dataset of cross-sectional assessments in age-matched TD children were selected from the established reference database at the CMAL-Pellenberg. This database included TD children with no history of neurological, neuromuscular or orthopedic disorders. Criteria for selecting the TD children for the current study were: (a) aged between 6 months and 9 years old and (b) assessments collected with similar measurement equipment and protocol as applied for the SCP participants. The study protocol was approved by the Ethical Commission UZ/KU Leuven, Leuven, Belgium (S59945, S62187, S62645) and registered (NCT05197764). Written informed consent was obtained for all participants.

### 2.2. Measurements

The three-dimensional freehand ultrasound (3DfUS), combining a conventional 2D US device and a motion tracking system, was used. This technique has proven to be valid and reliable for measuring healthy and pathological muscles, even from early ages [13,28,29,30]. The US device settings were kept constant throughout the study period. Details on the 3DfUS equipment and measurement protocol are given in previous studies [19,20]. While positioned in prone, the most affected leg was measured in all SCP participants, whereas a random leg was selected for TD children. The most affected leg was defined according to the MG spasticity and ankle joint range of motion (ROM) measured during a standardized clinical examination (Modified Ashworth Scale and goniometry, respectively) at the time of the baseline assessment [31]. The assessed leg remained the same for all follow-up assessments. Three experienced researchers were involved in the assessments with only 2 researchers for the same participant. Two of these researchers undertook the data processing, one of whom performed the data analysis. During the processing of the US images, the researchers were blinded to the GMFCS levels and timing of the assessment. All involved researchers were extensively trained in the CMAL research group and followed predefined guidelines and regular meetings to maximize similarities in the prescribed acquisition and processing workflow [32]. The inter-rater and intra-rater inter-session reliability was previously demonstrated with an intra-class correlation of minimal 0.95 [33]. Furthermore, the Portico (i.e., concave-shaped holder combined with concave gel pad) was used to minimize inter-acquire differences in probe pressure [34]. STRADWIN software (Version 6.0, Mechanical Engineering, Cambridge, UK) was used for both data acquisition and processing. MV (in mL) was defined by manually drawing muscle segmentations alongside the inner muscle border, starting from the medial femoral condyle until the last image of the muscle belly before approaching the muscle tendon junction. Subsequently, a cubic planimetry technique was applied for interpolation of the defined segmentations [35]. The mid-belly segmented anatomical CSA was extracted and is further referred to as the CSA (in mm^2^). The ML of the MG muscle (in mm) was defined as the Euclidean distance between the muscle origin at the most superficial aspect of the medial femoral condyle and the muscle tendon junction. Accounting for inter-individual anthropometric variability and in line with previous studies, ratio scaling was applied for which the MV was normalized to the product of body mass and length (nMV, mL/(kg·m)), CSA to body mass (nCSA, mm^2^/kg) and ML to body length (nML, %) [14,36].

### 2.3. Statistical Analysis

Participant characteristics and muscle outcomes at the baseline assessment were summarized by descriptive statistics reporting frequencies and median (interquartile ranges, IQ1-IQ3). Non-normality was confirmed with Shapiro–Wilk’s test. Two subgroups considering the motor impairments were used: subgroup 1, GMFCS level I and subgroup 2, GMFCS level II & III. To model the change in muscle morphology with respect to age, non-linear mixed models were used [37,38]. These models accommodate (a) unbalanced longitudinal datasets, resulting from variable spacing of follow-up intervals and different numbers of follow-up assessments among the children with SCP, and (b) the correlation between the repeated assessments taking the variance between and within the children with SCP into account. Fixed effects were (a) the age (expressed in years) at the time of assessments within each child, representing the time-effect and (b) covariate, representing GMFCS with its specific subgroups to describe within-group and compare between-group changes in the muscle growth. Random intercepts were added to model the variability in the starting point between the children with SCP. To capture the changes in the trajectory of morphological muscle growth with age, piecewise models were allowed. The following workflow was used to construct these models: first, the average trend (i.e., mean structure) was explored by performing Loess regressions and by plotting the observed individual longitudinal trajectories. These explorations suggested piecewise trends, i.e., constatation of linear trends interrupted by breaking points. Second, residual trajectories calculated from the Loess regressions and the observed variance function (i.e., the change of the squared residuals over time) were explored, to define the random-effect structure. A random intercept was selected based on the exploration closely approaching a constant variance. Estimated starting values from the graphical explorations were used in the logistic piecewise models to obtain the predicted intercept, regression coefficients, breakpoints at specific ages, variance of the random intercept and measurement error. 

The observed morphological outcomes included the MV, nMV, CSA, nCSA, ML and nML. The model with the *j*th observation in child *i* for GMFCS group *g (*GMFCS I *=* _I_ and GMFCS II–III = _II–III_ to estimate the response was described as follows:
$$\begin{array}{l} {\mathrm{Respons}}_{\mathrm{ijg}}=(\alpha_{0}g+a1ig)+\beta_{1}g*ageijg+\varepsilon_{(1)ijg}\;\;\;\;\;\;\;\;\;\;\;\;\;\;\;\;\;\;\;\;\;\;\;\;\;\;\;\;\mathrm{if}\;\mathrm{age}\;<c_{1}\\ {\mathrm{Respons}}_{\mathrm{ijg}}=(\alpha_{0}g+a1ig)+c_{1}g*(\beta_{1}g-\beta_{2}g)+\beta_{2}g*ageijg+\varepsilon_{(1)ijg}\;\;\;\;\;\;\;\;\;\;\;\;\;\;\;\mathrm{if}\;c_{1}\;\leq \mathrm{age}\;<c_{2}\\ {\mathrm{Respons}}_{\mathrm{ijg}}=(\alpha_{0}g+a1ig)+c_{1}g*(\beta_{1}g-\beta_{2}g)+c_{2}g*(\beta_{2}g-\beta_{3}g)+\beta_{3}g*ageijg+\varepsilon_{(1)ijg}\;\;\mathrm{if}\;\mathrm{age}\;\geq c_{2} \end{array}$$

With α_0_ = intercept; a1i = random intercept; c_1_ = first breakpoint; c_2_ = second breakpoint; β_1_ = regression coefficient of the slope before first breakpoint; β_2_ = regression coefficient of the slope after first breakpoint and before second breakpoint; β_3_ = regression coefficient of the slope after second breakpoint; ε_(1)ijg_ = measurement error.

An F-test was used to test (a) if the slopes and the breakpoints differ from zero, and (b) if the slopes before and after the breakpoint differ from each other, confirming the non-linear trajectory. After formulating the piecewise regression model, so-called empirical Bayes estimates were calculated and used to assess the potential presence of outliers, i.e., patients with an “exceptional” starting point and evolution over time. Further, the F-test was used to compare the regression coefficients and breakpoints within and between the GMFCS groups. *p*-values were set at <0.05. 

To enhance interpretation of the intercept, the age at all assessments was subtracted by 2 years, and therefore, the intercept represented the response’s value at the age of 2 years. The regression coefficients of the slope represent the magnitude of response in morphological muscle growth per year. The observed individual outcomes, observed individual trajectories, predicted individual trajectories and predicted average trajectory per morphological muscle outcome were plotted. The observed individual trajectories were also presented per GMFCS level, providing descriptive results per GMFCS level (Appendix A). The individual observed outcomes of 102 TD children were visualized with grey dots combined with boxplots for median (interquartile ranges) per 1-year age groups, illustrating a cross-sectional reference dataset compared to the longitudinal trajectories of the SCP children. For the TD children, the median cross-sectional growth rate for a specific age range was calculated as the median of the muscle outcome divided by the median age. Illustrations of morphological growth trajectories in each GMFCS group of the children with SCP compared to a retrospective dataset of cross-sectional assessments in TD children were added in Appendix A and Appendix A. All analyses were performed in SAS^®^ (Statistical Analysis Software version 9.4, SAS Institute Inc., Cary, NC, USA).

## 3. Results

Patient characteristics at the time of baseline assessment are summarized in Table 1. At the end of the follow-up, muscle data was collected until 10.3 years for GMFCS I and 11.1 years for GMFCS II–III (Figure 1). Only nine children changed from GMFCS level during the follow-up for which the re-classification was performed around 2–3 years of age. The study sample received standardized clinical care such as regular physical therapy, orthotic devices and medical and orthopedic services (e.g., serial casting and/or BoNT-A injections when indicated) within a multidisciplinary clinical setting (Appendix A). The results of the piecewise models for the subgroups in GMFCS levels are summarized in Table 2, Figure 2 and Appendix A. The comparisons of slopes and breakpoints between the groups are presented in Table 3.

For the GMFCS Level I group, the MV increased with 12.8 mL/year (β_1_, *p* < 0.0001) up to 2.1 years of age (c_1_). After these infant ages, an MV increase of 5.7 mL/year was found until the age of 7.8 years (β_2_, *p* < 0.0001 and c_2_, respectively). From this second breakpoint at these older ages, a growth rate of 3.1 mL/year was found (β_3_, *p* = 0.0024) which was significantly lower compared to growth rates for the younger ages (β_1_ vs. β_3_ and β_2_ vs. β_3_, *p* < 0.05, Appendix A). On the other hand, after an increase in nMV of 0.83 mL/kg·m per year until 2.1 years, the nMV showed a non-significant rate of −0.03 mL/kg·m per year until the age of 8.0 years (β_2_, *p* = 0.0933 and c_2_). After this second breakpoint, a rate of −0.12 mL/kg·m per year was found (β_3_, *p* < 0.0001) for which this linear trend showed significantly more decline compared to the muscle growth rates before 2 years and between 2 and 8 years old (β_1_ vs. β_3_ and β_2_ vs. β_3_, *p* < 0.05, Appendix A). Similar to the growth rate in MV, the trajectory of CSA growth showed two breakpoints (age 2.2 years and 6.7 years, respectively) with only in the first and second linear trend, a significant yearly increase in CSA (β_1_ = 158.4 mm^2^/year, *p* < 0.0001; β_2_ = 37.0 mm^2^/year, *p* < 0.0001, respectively). The nCSA trajectory showed after an increase of 8.88 mm^2^/kg per year only 1 breakpoint at the age of 2.1 years followed by decrease in nCSA per year (c_1_ and β_2_= −0.96 mm^2^/kg per year, *p* < 0.0001, respectively).

After a significant MV increase with a rate of 4.5 mL/year (β_1_, *p* < 0.0001), children in the GMFCS level II–III group showed only a breakpoint at the age of 9.1 years (c_1_) which was followed by a rate of −0.8 mL/year (β_2_, *p* = 0.7154). The nMV showed increases of 0.28 mL/kg·m per year until the breakpoint at the age of 2.7 years (c_1_) and followed by decreases of 0.07 mL/kg·m per year (β_2_, *p* = 0.0003). The trajectories for absolute and normalized CSA were modelled with two breakpoints, around approximately 2 and 9 years of age. During the (pre)school ages, nCSA decreased with significantly higher decline after the age of 9.1 years (β_2_ vs. β_3_, *p* = 0.0424, Appendix A).

Before the age of 2 years, children with GMFCS level I showed significantly higher yearly increases for both absolute and normalized MV and CSA compared to level II–III (β_1, GMFCS-I_ vs. β_1, GMFCS-II–III_, Table 3, *p* < 0.05). After the age of 2 years, absolute MV and CSA were significantly increasing in both GMFCS subgroups until the age of 6–9 years old. However, GMFCS level I tended to have a higher absolute MV growth rate compared to GMFCS level II–III (β_2, GMFCS-I_ vs. β_1, GMFCS-II–III_ =1.16 mL/year, *p* = 0.0712). Despite the significant earlier second breakpoint for GMFCS levels I (Δ1.31 years for MV, *p* = 0.0326 and Δ2.66 years for CSA, *p* = 0.0033), no significant differences were found in the MV and CSA growth rate at these oldest ages between the GMFCS subgroups (β_3, GMFCS-I_ vs. β_2, GMFCS-II–IIII_, *p* = 0.1069 and β_3, GMFCS-I_ vs. β_3, GMFCS-II–III_, *p* = 0.0915, respectively). The trajectory for nMV and nCSA showed decreased rates from the age of 2 years in both GMFCS groups, with a significant higher decline in nCSA in children with GMFCS level II–III after the age of 6–9 years compared to level I (β_2, GMFCS-I_ vs.β_3, GMFCS-II–III_, *p* = 0.0334). 

Absolute ML increased with age, with significantly lower ML growth rate after the age of 5.11 years in the GMFCS level I and after the age of 2.1 years in GMFCS levels II–III (β_1_ vs. β_2_, *p* < 0.05). Normalized ML showed significant different rates with increasing age between the GMFCS groups, for which the GMFCS levels I increased with 0.33%/year until 4.88 years (β_1_, *p* = 0.0125) followed by a rate of −0.10%/year (β_2_, *p* = 0.1977), whereas the levels II–III increased in nML with 0.12% per year from infancy to school ages, without a breakpoint (β_1_, *p* = 0.0096).

## 4. Discussion

In this prospective longitudinal follow-up between 6 months and 11 years of age, the trajectory of morphological muscle growth for children with SCP was observed as piecewise profiles with increasing age, indicating linear trends interrupted with breakpoints. Therefore, our hypothesis of gradual increases in muscle morphology with increasing age could only be partially accepted as expressed by the changes in observed yearly muscle growth rates before and after the breakpoints. Indeed, MV and CSA significantly increased before the age of 2 years but this was followed by slower increases until the age of 6–7 years for GMFCS level I and 9 years for GMFCS level II–III. Normalized MV and CSA also increased during infancy but showed already reduced growth rates after the age of 2 years. Furthermore, from 8–9 years of age, both the absolute and normalized MV and CSA outcomes reduced per year, which is in contrast with the hypothesis of a linear trajectory of muscle growth with increasing age. Absolute ML increased over the entire age range with a breakpoint at the age of 5 years for GMFCS level I and 2 years for GMFCS level II–III, whereas nML showed less increase after this age for GMFCS level I. Lower muscle growth rates with increasing age in children of the higher GMFCS levels was hypothesized and was only confirmed for the absolute morphological growth before the age of 2 years and in teenagers. Next, the normalized MV and CSA outcomes reduced per year with a more pronounced decrease in CSA normalized to changes in body dimensions for GMFCS level II–III. Further, ML trajectories were different between the GMFCS levels and more specific, after normalization for skeletal growth.

These results revealed higher rate of muscle growth in early years of life, with lower growth rates for children with GMFCS level II–III compared to GMFCS level I. During these youngest ages, the current MV increased with 12.8 mL per year for GMFCS level I, which was slightly higher compared to the median growth rate of 10.3 mL per year. This growth rate is computed by dividing the average MV of 15.45 mL to the time period of 1.5 years to represent the 6 months to 2 years old TD children of our retrospective cross-sectional database (Appendix A and Appendix A). Only one previous study reported a cross-sectional growth rate for 8 months to 5-years-old children, with 8.16 mL/year for TD and 3.84 mL/year for CP children [13]. The latter is comparable to the growth rate of 4.5 mL/year in the current GMFCS level II–III children suggesting an early onset of more pronounced growth deficits in children with more severe impairments. Furthermore, the nMV rates of 0.83 mL/(kg·m) and 0.28 mL/(kg·m) per year were lower than our cross-sectional TD growth rate of 1.17 mL/(kg·m) per year. No growth rates for nMV during early ages have been previously reported. Nevertheless, growth rate calculated on cross-sectional datasets assumed linear growth, whereas the current results revealed non-linear trajectories of muscle growth with increasing age. However, caution is warranted with the interpretation of observed increases in normalized muscle data. In the case of optimal normalization for changes in body sizes, growth rates close to zero are expected, representing harmonization between muscle and anthropometric growth [39]. The validity of ratio-scaling in growing TD children, over a wide age range, has not yet been defined. The body composition in children is assumed to change with less amount of fat mass after the age of 2 years. Hence, normalization by taking body mass into account is considered invalid during infancy [40]. Further, significantly lower intercepts for nMV, nCSA and nML indicated that children with less motor abilities develop more muscle pathology already early in life (Table 3). These observations suggest early mechanisms hampering the muscle growth, such as the contribution of neural alterations as a result of the brain lesion as well as altered patterns of muscle use. However, limited muscle data on TD infants is available to compare the current suggestion of an early onset of muscle alterations, especially in GMFCS levels II–III. Further research also involving longitudinal assessments of TD infants is required to explore the impact of normalization techniques and provide valid references to investigate the onset and development of muscle alterations in infants with SCP.

Interestingly, from 2 years of age, similar muscle growth trajectories in (normalized) MV and CSA for the GMFCS subgroups were found. The average MV rates of 5.1 mL/year and 4.5 mL/year in GMFCS level I and level II–III, respectively, were comparable to earlier reported MV growth rates based on 6 months and 12 months follow-up studies in 2–5 years old children with SCP (6.0 mL/year and 6.6 mL/year, respectively) [14,41]. Furthermore, hampered morphological muscle growth is more clearly highlighted by investigating the changes in normalized MV and CSA compared to the investigation of the absolute parameters. It should be noted that altered anthropometric growth indicated by shorter body length and lower body mass, have already been shown in children with SCP compared to TD peers [42]. These growth deficits were found to be associated with increasing age, as well as with bilateral involvement and more severe gross motor impairments [43]. Although anthropometric growth in children with minor motor impairments is close to age-matched TD peers [42], the significant negative normalized muscle growth rates observed in the current study for the GMFCS level I group showed that the muscle growth was not in accordance with the skeletal growth. Since muscle morphological growth is assumed to be triggered by anthropometric growth and patterns of muscle use which might be defined by the gross motor functional abilities, the observed early presence of muscle pathology for GMFCS level II–III and increasing deficits after accounting for normalization to changes in body dimensions suggest that pathology-related biomechanical triggers contribute to hampered muscle growth.

From the age of 6–9 years, hampered muscle growth was presented in both GMFCS groups. Despite the lower number of data points at these older ages for children with GMFCS levels I, the level of gross motor mobility in daily life is suggested to further contribute to hampered growth as shown by the tendency of lower CSA growth and significantly lower nCSA growth for GMFCS level II–III compared to level I. These findings are in line with cross-sectional observations of significantly smaller MV and CSA in GMFCS levels II compared to level I for 5–12 years old children [18]. However, the current data set did not further distinguish the growth trajectories between GMFCS levels II and III for which the function level and severity of neuromotor impairments could be quite different. While the underlying mechanisms of these changes in trajectory between the GMFCS levels are currently unclear, they might be attributed to less weight bearing activities, reduced levels of physical activity, higher incidence of nutritional problems and increased (secondary) musculoskeletal alterations, especially from 6 to 9 years [11].

The current findings also suggested a different trajectory for the growth in cross-sectional and longitudinal dimension of the MG muscle, i.e., changes in CSA and ML, respectively. For CSA, a growth trajectory with an early breakpoint at 2 years of age was observed in both GMFCS groups, whereas the trajectory of ML showed a breakpoint at the age of 4–5 years in GMFCS level I and at the age of 2 years in GMFCS level II–III. The trajectories in CSA are aligned to the changes in MV, suggesting that alterations in cross-sectional MG muscle growth may be associated to the change in overall MG muscle size. This is in line with previous findings indicating that reduction in physiological CSA, a determinant for force generation, is determined by reduced MV rather than by changes in fascicle length [44,45]. Hence, clinical utility may be found in assessing only the mid-belly anatomical CSA as primary outcome for monitoring the muscle status and treatment follow-up.

With increasing age, evolution in neuromotor symptoms can be expected due to the natural history of SCP in children. Interestingly, breakpoint models describing changes in development at specific ages were already found in longitudinal follow-up studies for spasticity and lower limb ROM [5,6,46]. These previous studies showed that spasticity increases to the age of 4 years, followed by a decrease each year until the age of 12 year [6]. Decreasing ankle joint ROM was reported up to 5 years of age, followed by further decrease with age in GMFCS level I and II, while levels III showed increased ROM [5]. The current study showed similar non-linear models for the ML outcomes in GMFCS level I with a breakpoint at the age of 5.11 years. The nML changed from 0.33% per year to −0.10% per year for GMFCS level I, whereas a constant increase of 0.12%/year was presented for GMFCS level II–III. Therefore, the restricted ROM for GMFCS level I might result from reduced growth of muscle length. Furthermore, these similarities in longitudinal observations of neuromotor impairment and muscle development, including the breakpoint at 5 years, support the hypothesis that formation of contractures is triggered by different factors, of which spasticity is probably not the dominant one [46,47].

Over the last years, 3DfUS has already been extensively used mainly to describe muscle deficits in children with SCP compared to TD children [48]. These cross-sectional investigations included diverse age ranges and GMFCS levels resulting in variable results of deficits (for example 22% for the age range 2–5 years and 41% for 5–12 years old children) [16,19]. However, age-specific observations of hampered muscle growth for SCP children were missing in these previous studies. Our findings, derived from longitudinal models, emphasized the need for repeated assessments to accurately delineate muscle growth. Furthermore, the current muscle growth trajectories can be used to monitor the status of muscle pathology in individual children with SCP. The current results may support the clinical decision making of therapy selection, goals and planning at specific ages and for each GMFCS level with the aim of stimulating the muscle growth. For example, the application of strength training, considering the appropriate age and cognitive functioning, might be beneficial for maintaining the muscle size relative to the skeletal growth and might be further combined with age- and child-specific prescription of protein intake to increase the muscle size [49,50,51]. Beside interventions, the current observation of early muscle alterations for higher GMFCS levels and the breakpoint in growth trajectory at a very young age highlight the opportunity for prevention strategies aiming to maintain muscle size and lengths comparable to TD children and preserve muscle growth during childhood, e.g., intensive physical therapy with stimulation of lower leg movements and mechanical loading and a nutritional plan. Previous studies focusing already on the impact of BoNT-A injections, a frequently applied tone-reducing treatment, demonstrated hampered muscle growth in response to the first BoNT-A treatment [14,52]. This post-treatment effect on muscle growth was only assessed after 6 months for which the interference of BoNT-A injection with the trajectory of muscle changes is yet unclear. Future intervention studies could use the currently modelled muscle growth trajectories as a reference to assess the effect of treatment on muscle morphology and could fine tune the timing of treatment in relation to the potential hampered growth.

This study has some limitations to consider. First, the age-matched TD data could not be included in the models due to the lack of longitudinal assessments. Yet, visual inspection of cross-sectionally measured typical muscle morphology allowed one to judge the overall level of alterations in the data of children with SCP (Appendix A). We included children aged between 6 months and 9 years at baseline resulting in limited available data points at the beginning and end of the age range. More longitudinal data before and after this range is relevant to further enrich the muscle growth trajectories during the entire childhood. Combined with longitudinal data of TD children, a focused analysis of the muscle growth before 2 years of age and after 6 years of age would be interesting for future research. Second, this unique longitudinal database is limited to provide trajectories of medial gastrocnemius muscle growth for the specific diversity in SCP phenotypes such as the level of motor abilities and the topographic classes. In this study, only the GMFCS levels representing ambulant children with SCP were included. The GMFCS level II and III were merged to ensure sufficient power for the data analysis and to provide as much homogeneity as possible. By combining the data of children with GMFCS II and III across the broad age range from 6 months to 11 years, children who function at level II and use gait aids at younger ages when learning to walk, or walking longer distances, are included. This is distinct from children functioning at level I, who walk independently without the need for a gait aid. The current dataset was, however, limited in data points after the age of 6 years to describe the trajectories of muscle growth per GMFCS level. We only descriptively explored the muscle growth per GMFCS level with the individual observed profiles (Appendix A), indicating similar trajectories in morphological muscle growth. Nevertheless, it is important to further consider the potential heterogeneity in muscle growth based on the specificity in gross motor abilities and limitations through daily life per GMFCS level. Future studies should aim for more participants after the age of 6 years and with equally distributed number of children over each of the three ambulatory GMFCS levels in order to distinguish the models for muscle growth according to the GMFCS level. Next, the muscle growth associated with the SCP topography i.e., unilateral versus bilateral SCP was also not specifically investigated. However, further sub-analyses are important considerations due to the unbalanced number of participants between the topographic classes. Anthropometric growth, lower limb strength and gait were found to be less involved in unilateral compared to bilateral SCP, whereas only one investigation showed more muscle growth deficits for this SCP motor type [53,54]. These findings suggest the need to describe, in future research, separate muscle growth trajectories for the children with uni- and bilateral SCP in interaction with GMFCS level. As a first exploration, the observed individual trajectories per motor subtype for the current GMFCS subgroups are provided in Figure 3. Visual inspection suggests less muscle growth in children with bilateral SCP, especially for GMFCS level II–III of which 65% of the children were bilaterally involved.

Further, each muscle assessment during the follow-up was not accompanied with evaluation of motor impairments and functioning such as assessment of strength, spasticity and selective motor control as well as the gross motor function measure (GMFM). These clinical results combined with muscle growth data could create a comprehensive monitoring of SCP muscle pathology, which may help to understand the underlying mechanisms of altered muscle growth and eventually help to improve patient-specific treatment management. Of note, in the current study, children who received BoNT-treatment were allowed for further follow-up in cases when the BoNT-A treatment occurred minimal 10 months prior to the assessment. The current database included 24 children who received BoNT-A injection during the follow-up time. Our previous research highlighted that the muscle recovery is still ongoing at 6 months post BoNT-A injections [14]. Despite the 10-month interval between the BoNT-A treatment and muscle morphology assessments, which is already longer than the frequently used criteria of an interval of 6 months, there is no guarantee for full recovery of muscle growth at the follow-up assessment. Future studies should investigate the prolonged impact of treatments. Long-term follow-up assessments are essential to further understand if BoNT-A treatment is a confounding factor on the currently established trajectories of muscle growth.

## 5. Conclusions

To our knowledge, this was the first study that performed repeated muscle assessments over 2 years in an extended group of growing children with SCP and demonstrated piecewise model for MG muscle growth. The study sample is believed to be representative of the population of children receiving standardized clinical care in Belgium (Appendix A). After high rates of muscle growth during the first 2 years of life, the trajectory of absolute MV and CSA changed, resulting in a slower gradual growth until 6–9 years, and subsequent reduction in muscle size growth after 9 years. With increasing age, normalized MV and mid-belly CSA represented a decrease in muscle morphology after approximately 2 years of age. Children with GMFCS level II–III showed more muscle growth reduction compared to children with level I. This lower growth rate started at a very young age. After infancy until approximately 9 years of age, the trajectory of muscle growth did not differ greatly between GMFCS levels. These longitudinal trajectories could be used to monitor the SCP muscle pathology during childhood and to optimize treatment planning and goals aiming to stimulate muscle growth.

## Figures and Tables

**Figure 1 jcm-12-01564-f001:**
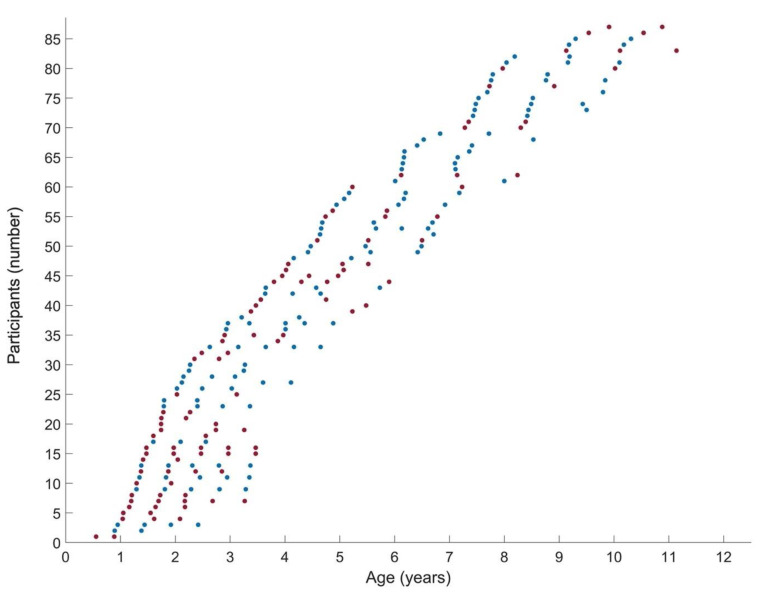
Representation of the number of the performed longitudinal assessments at the different ages for the group of children with spastic cerebral palsy (SCP) (*n* = 87) of gross motor function classification system level I (blue) versus levels II–III (red). The *x*-axis represents the ages at the time of the repeated assessment. On the *y*-axis, each participant is represented on one line, with each dot representing the performed assessments. The children were further ordered with increasing age. The group of GMFCS level I included 47 children with a total of 130 assessments (average (range) of 2.8 (2–5) assessments per child and average time of 10 months (4–25 months) between the assessments. The group of GMFCS level II–III included 40 children with a total of 104 assessments (average (range) of 2.6 (2–5) assessments per child and average time of 9 months (4–25 months) between the assessments).

**Figure 2 jcm-12-01564-f002:**
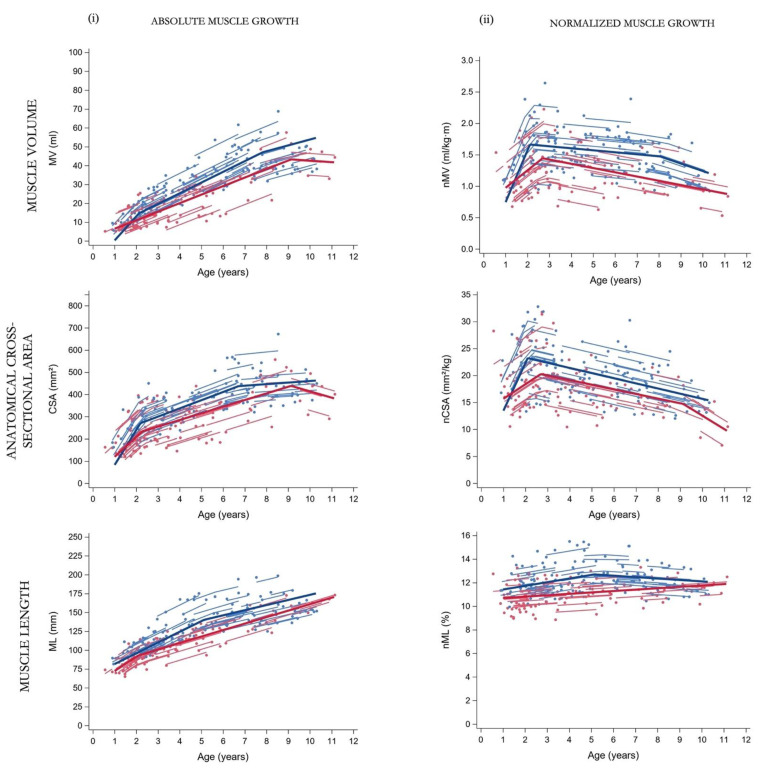
Predicted average trajectories (bold solid line) for the (**i**) absolute and (**ii**) normalized muscle morphology in SCP children with GMFCS level I (blue) and GMFCS level II–III (red). Observed outcomes (dots) and individual predicted trajectories (lines) are presented. SCP, spastic cerebral palsy; GMFCS, gross motor function classification system; MV, muscle volume; ml, milliliter; CSA, anatomical cross-sectional area; mm, millimeter; ML, muscle belly length; n, normalized; kg, kilogram; m, meters.

**Figure 3 jcm-12-01564-f003:**
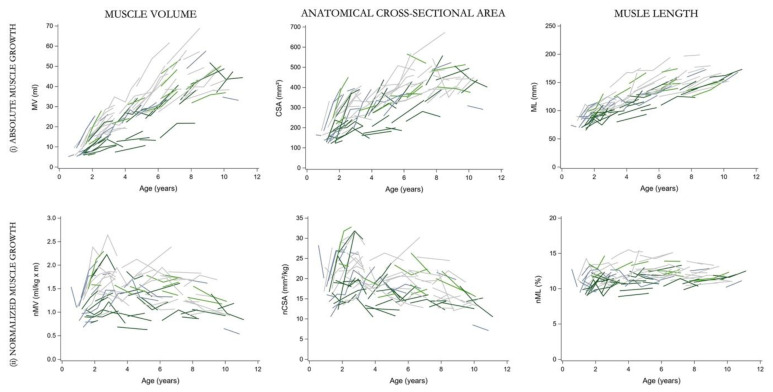
Observed individual trajectories for the (**i**) absolute and (**ii**) normalized muscle morphology for children with unilateral SCP (*n* = 51) (grey) and children with bilateral SCP (*n* = 36) (green), in GMFCS level I (light colors) and GMFCS level II–III (dark colors) group. GMFCS, gross motor function classification system; SCP, spastic cerebral palsy; n, number; MV, muscle volume; ml, milliliter; CSA, anatomical cross-sectional area; mm, millimeter; ML, muscle belly length; n, normalized.

**Table 1 jcm-12-01564-t001:** Participant characteristics of the children with SCP at baseline assessment.

Group	GMFCS I	GMFCS II–III
Participants	47	40
Observations	130	104
Sex (male/female)	25/22	21/19
GMFCS level (I/II/III)	47/0/0	0/22/18
Involvement (unilateral/bilateral)	37/10	14 (II, = 10 & III = 4)/26 (II = 12 & III = 14)
History of BoNT-A injection	11	7
Antropometric & muscle outcomes		
Age (y)	4.66 (2.25–6.83)	2.88 (1.47–4.81)
Body mass (kg)	18.0 (13.0–22.0)	12.5 (10.0–16.4)
Body length (cm)	104.7 (88.5–121.1)	90.0 (77.8–106.0)
MV (mL)	28.2 (16.7–38.4)	13.6 (7.4–25.5)
CSA (mm^2^)	335.6 (260.1–392.1)	205.5 (162.3–285.8)
ML (mm)	127.1 (99.2–144.9)	96.0 (82.8–127.1)
nMV (mL/kg·m)	1.45 (1.26–1.58)	1.12 (0.92–1.39)
nCSA (mm^2^/kg)	19.4 (16.1–22.1)	16.1 (13.8–19.0)
nML (%)	11.7 (11.3–12.6)	10.9 (9.8–11.9)

The frequencies are presented for the general characteristics. Anthropometric and muscle data are presented as median (interquartile 1–interquartile 3). SCP, spastic cerebral palsy; GMFCS, gross motor function classification system; BoNT-A, botulinum neurotoxin type A; y, years; kg, kilogram; cm, centimeter; MV, muscle volume; ml, milliliter; CSA, anatomical cross-sectional area; mm, millimeter; ML, muscle belly length; n, normalized; kg, kilogram; m, meter.

**Table 2 jcm-12-01564-t002:** Fixed effects of the piecewise model for muscle morphology in GMFCS subgroups (level I, *n* = 47; levels II–III, *n* = 40).

		Intercept	Regression Coefficients (β) and Breakpoints (c)				
Outcome	Participants	α_0_	β_1_ (CI) *p*-*Value*	c_1_	β_2_ (CI) *p*-*Value*	c_2_	β_3_ (CI) *p*-*Value*
MV	GMFCS I	13.2	12.8 (8.4–17.1)	2.12 *	5.7 (4.6–6.7)	7.82 *	3.1 (1.1–5.1)
			** *<0.0001* **		** *<0.0001* **		** *0.0024* **
		α_0_	β_1_ (CI) *p*-*value*			c_1_	β_2_ (CI) *p*-*value*
	GMFCS	11.1 *	4.5 (3.8–5.2)			9.13 *	−0.8 (−5.1–3.5)
	II–III		** *<0.0001* **				*0.7154*
		α_0_	β_1_ (CI) *p*-*value*	c_1_	β_2_ (CI) *p*-*value*	c_2_	β_3_ (CI) *p*-*value*
nMV	GMFCS I	1.57*	0.83 (0.55–1.11)	2.11 *	−0.03 (−0.07–0.01)	8.04 *	−0.12 (−0.18–−0.06)
			** *<0.0001* **		*0.0933*		** *<0.0001* **
		α_0_	β_1_ (CI) *p*-*value*	c_1_	β_2_ (CI) *p*-*value*		
	GMFCS	1.25 *	0.28 (0.14–0.42)	2.69 *	−0.07 (−0.10–−0.03)		
	II–III		** *<0.0001* **		** *0.0003* **		
		α_0_	β_1_ (CI) *p*-*value*	c_1_	β_2_ (CI) *p*-*value*	c_2_	β_3_ (CI) *p*-*value*
CSA	GMFCS I	241.1 *	158.4 (115.6–201.2)	2.18 *	37.0 (23.1–50.9)	6.71 *	6.96 (−8.35–22.3)
			** *<0.0001* **		** *<0.0001* **		*0.3688*
	GMFCS	209.5 *	88.6 (0.99–1.38)	2.27	29.8 (4.62–6.80)	9.13 *	−28.3 (−66.8–10.3)
	II–III		** *<0.0001* **		** *<0.0001* **		*0.1483*
		α_0_	β_1_ (CI) *p*-*value*	c_1_	β_2_ (CI) *p*-*value*		
nCSA	GMFCS I	22.4 *	8.88 (5.42–12.3)	2.09 *	−0.96 (−1.38–−0.54)		
			** *<0.0001* **		** *<0.0001* **		
		α_0_	β_1_ (CI) *p*-*value*	c_1_	β_2_ (CI) *p*-*value*	c_2_	β_3_ (CI) *p*-*value*
	GMFCS	18.4 *	2.70 (0.21–5.19)	2.70 *	−0.86 (−3.89–−1.14)	9.13 *	−2.52 (−3.89–−1.14)
	II–III		** *0.0336* **		** *0.0138* **		** *0.0005* **
		α_0_	β_1_ (CI) *p*-*value*	c_1_	β_2_ (CI) *p*-*value*		
ML	GMFCS I	95.9 *	14.2 (12.2–16.3)	5.11 *	6.80 (4.57–9.02)		
			** *<0.0001* **		** *<0.0001* **		
	GMFCS	91.3 *	18.8 (11.2–26.5)	2.10 *	8.59 (7.56–9.62)		
	II–III		** *<0.0001* **		** *<0.0001* **		
nML	GMFCS I	11.7 *	0.33 (0.07–0.58)	5.13 *	−0.10 (−0.26–0.06)		
			** *0.0125* **		*0.1977*		
				β_1_ (CI) *p*-*value*			
	GMFCS	10.8 *	0.12 (0.03–0.21)				
	II–III		** *0.0096* **				

The asterisks (*) indicate significance level at *p* < 0.05 for α and c. *p*-values in bold indicate significance level at *p* < 0.05. The following symbols represent: α = outcome at the age of 2 years, β = rate of change in muscle outcome per year, and c = age (years) of the breakpoint. The numbers (in subscript) associated with symbols β and c refer to the order of the observed growth ratio and breakpoint, respectively, e.g., β_1_ as the first ratio before the c_1_, the first breakpoint. SCP, spastic cerebral palsy; GMFCS, gross motor function classification system; n, number; CI 95% confidence interval; MV, muscle volume; n, normalized; CSA, anatomical cross-sectional area; ML, muscle belly length.

**Table 3 jcm-12-01564-t003:** Results of the comparison in slopes and breakpoints between the GMFCS subgroups.

Outcome	GMFCS I vs. II–III	Δ	*p*-Value
MV	β_1, I_ vs. β_1, II–III_	8.24	**0.0003**
	β_2, I_ vs. β_1, II–III_	1.16	0.0712
	β_3, I_ vs. β_2, II–III_	3.91	0.1069
	c_2, I_ vs. c_1, II–III_	1.31	**0.0326**
nMV	β_1, I_ vs. β_1, II–III_	0.55	**0.0007**
	β_2, I_ vs. β_2 II–III_	0.04	0.1710
	β_3, I_ vs. β_2, II–III_	0.05	0.1272
	c_1, I_ vs. c_1, II–III_	0.58	**0.0040**
	α_1, I_ vs. α_1, II–III_	0.32	**0.0056**
CSA	β_1, I_ vs. β_1, II–III_	69.8	**0.0112**
	β_2, I_ vs. β_2, II–III_	7.17	0.4328
	β_3, I_ vs. β_3, II–III_	35.2	0.0915
	c_1, I_ vs. c_1, II–III_	0.09	0.3783
	c_2, I_ vs. c_2, II–III_	2.66	**0.0033**
nCSA	β_1, I_ vs. β_1, II–III_	6.18	**0.0039**
	β_2, I_ vs. β_2, II–III_	−0.09	0.8168
	β_2, I_ vs. β_3, II–III_	1.56	**0.0334**
	c_1, I_ vs. c_1, II–III_	0.61	0.1152
	α_0, I_ vs. α_0, II–III_	3.97	**0.0063**
ML	β_1, I_ vs. β_1, II–III_	−4.61	0.2482
	β_2, I_ vs. β_2, II–III_	−1.79	0.1384
	c_1, I_ vs. c_1, II–III_	−3.01	**<0.001**
nML	β_1, I_ vs. β_1, II–III_	0.21	0.1311
	β_2, I_ vs. β_1, II–III_	−0.22	**0.0146**
	α_0, I_ vs. α_0., II–III_	0.91	**0.0024**

*p*-values in bold indicate significance level at *p* < 0.05. The following symbols represent: α = outcome at the age of 2 years, β = rate of change in muscle outcome per year, c = age (years) of the breakpoint, Δ = difference score, I = GMFCS level I and II–III = GMFCS level II & III. The numbers (in subscript) associated with symbols β and c refer to the order of the observed growth ratio and breakpoint, respectively, e.g., β_1_ as the first ratio before the c_1_, the first breakpoint; GMFCS, gross motor function classification system; MV, muscle volume; n, normalized; CSA, anatomical cross-sectional area; ML, muscle belly length.

## Data Availability

The raw data supporting the conclusions of this article will be made available by the authors via an online repository, without undue reservation.

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
