# Peer review of "Morphological Medial Gastrocnemius Muscle Growth in Ambulant Children with Spastic Cerebral Palsy: A Prospective Longitudinal Study"

_jcm, 2023, doi:10.3390/jcm12041564_

Round 1

Reviewer 1 Report (Previous Reviewer 1)

This longitudinal dataset is an important contribution to the knowledge base regarding muscle morphology in children with spastic CP. My comments have been addressed. 

Author Response

Thank you for your reply and time to review this revised manuscript.

Reviewer 2 Report (Previous Reviewer 2)

The article got improved and it is making sense for the readers after corrections. All the best

Author Response

Thank you for your reply and time to review this revised manuscript.

Reviewer 3 Report (Previous Reviewer 3)

An excellent addition to the growing body of work in the area of muscle growth for children with cerebral palsy. The addition of 'break points' is novel, and offers clinical utility by dispelling the myth that growth is linear, better aligning with the typical way in which children present. Utilising this knowledge, the clinician may develop more targeted treatment goals and timing of interventions. The limitations are well addressed and the paper offers multiple viewpoints in considering your findings.

Thankyou for the opportunity to review this important work.

Author Response

We thank the reviewer for assessing the manuscript and providing this relevant feedback. We followed the suggestion to improve the readability. 

The changes in the manuscript are listed in red with the specific lines and indicated with track-changes in the manuscript_track changes.doc

Line 451-455:  By combining the data of children with GMFCS II and III with a  across the broad age range from 6 months to 11 years old children, the youngest children with GMFCS level II might still need gait aids as well when they learn to walk and/or when they walk for long distances while this might not be the case for children with GMFCS level I children who function at level II and use gait aids at younger ages when learning to walk, or walking longer distances, are included. This is distinct from children functioning at level I, who walk independently without the need for a gait aid.

This manuscript is a resubmission of an earlier submission. The following is a list of the peer review reports and author responses from that submission.

Round 1

Reviewer 1 Report

Overall comments:

Longitudinal data for muscle morphology in spastic cerebral palsy (CP) provides valuable information. As the population is heterogeneous, having multiple data points from the same individuals is important. There is fairly large group of children (n= 87) included in the study over a two-year period and this is a great opportunity to explore this data.

A problem with the data analysis was the choice to compare the data from participants at GMFCS levels I versus II and III. A scientific basis for this grouping was not provided. Rather, the rational was that the authors merged levels II and III to “ensure sufficient power for the data analysis” (p.15). Typically GMFCS levels I and II are grouped together as these levels indicate an ability to walk independently. GMFCS level III requires hand-held assistance. When the full spectrum of GMFCS levels are included, researchers often group those that are ambulatory (I, II and III) versus those who are not (IV and V). Additionally, the hemiplegic group would be included in levels I and II but it is unlikely that they would be level III. The data for the number of participants for bilateral and unilateral within the I/II group was not included. This choice prevents the reader from viewing all the available data, evaluating the results and believing some of the conclusions.

The authors describe differences between these groups as indicators motor ability but this classification system is limited to mobility levels. GMFCS is not a measure of motor function at the ankle joint. There is more weight bearing in the GMFCS level I and II groups and this the only clear distinction that might affect muscle morphology.

It would be more meaningful to present the data for all 3 GMFCS levels to the reader. If there is a scientific justification to combine GMFCS levels I/II based on the data, then it can provided. Figure 1 shows a concentration of data for children below 4- 5 years. Was there sufficient power to examine all three GMFCS levels below this age range? Figure S1 presented shows a nice contrast of all of the data compared to typically developing cross sectional data. Adding another column and showing the actual data for GMFCS II and III separately would be more meaningful than combining them.

Last session was used to determine GMFCS level. Did any participants change GMFCS levels

The Discussion is limited by the combining of these two GMFCS levels. Based on the literature, one would expect to see a difference between GMFCS level II and III.

Specific Comments

Abstract

P1 L30  The impact of “functional impairment” is not fully captured by an individual’s GMFCS Level, which is truly a mobility level of the entire body not the muscle itself. Mobility levels may affect altered muscle growth.

P1 L33  Repeated … sentence is confusing. Measured every 6 mos for 2 years.

Introduction

P2 L50 What is meant by “interact with” Is one driving the other etc.?

P2 L57  Not sure what is meant by “treated at the muscular level” botulinum toxin, stretching, therapy, bracing?

P2 L63 – reduce “by” instead of “with”?

P2 L76 – please provide a citation for each of the factors listed

P3 L87 – measure not an outcome. This entire sentence is long and repetitive…..

P3 L101 – confusing sentence. “prospective” but a “follow-up” instead of “longitudinal”

P3 L106 Appears to be a sample of convenience. Why do you state “to reduce heterogeneity”?

P3 L108 – Botulinum toxin is a common treatment that can affect morphology and is, therefore, a confounding factor. Would be cleaner to omit all botulinum toxin or keep tract to see if there was a difference in trajectories. Why choose botulinum toxin exclusion to be “10 months” prior?

P4 L132 - Were the researcher performing Imaging and quantification blinded to GMFCS level?

P4 L139  - Did the same researchers who performed imaging also perform morphology calculations. How many different people were involved in quantification. Is there any information about inter and intra rater reliability. This is not clear in 132 and 133?

P7 Fig. 1 – This figure legend is confusing. Longitudinal data sets or? children. Do you mean “of” etc.

P12 L286 –  Is piecewise models a finding? What did you find?

P12 L287 – What do you mean by increasing muscle morphology with increasing age. Be more specific to your actual findings.

P12 L304 – “Higher” not “highest”

P14 L369 – How does suboptimal moment arms related to overall growth trajectories?

P15 L411 – The “application of strength training after 2 years of age” is not developmentally appropriate. Alterations in diet should be child specific.

Author Response

Dear reviewer,

Please see the attachment. This report summarizes the authors’ replies to each comment or suggestion (marked in bold) of the reviewers by point-by-point answers (marked in blue font). Changes and adaptations in the manuscript are listed in red with the specific lines and indicated with track-changes in the manuscript_track changes.doc

Reviewer 2 Report

This is an excellent study done by the authors. However, it is a little complicated for the common reader to understand. I request authors simplify it at every stage, keeping an undergraduate or graduate student in their mind. I have provided minor comments at every stage to make the study more interesting, and authors can also put a little more effort into making it a successful publication.

Abstract:

Page 1, line number 42-44: Rather than providing future suggestions, make a stronger conclusion based on your study finding.  

Introduction:

Page 2, line 54: The reference for GMFCS was related to GMFCS- Expanded and Revised. Hence please make it clear whether you used GMFCS or GMFCS-E&R.

Page 2-3, lines 48-96: The whole introduction as a single continuous paragraph is difficult to understand.

Page 3, lines 94-96: It is clear by the previous researchers that children with spastic cerebral palsy have decreased muscle growth as per their age. However, your aim was to compare this muscle growth with age and GMFCS levels. Is there any other objective than this in your study.

Methods:

Page 3, line 102: It is unclear how many measurements each subject underwent.

Results

Page 6, table 1, line 220: Along with baseline measurements personally, as a consultant in pediatric physical therapy, I am interested to know the values of Body mass (kg), Body length (cm), MV (ml), CSA (mm2), ML (mm), nMV (ml/kgm), nCSA (mm2/kg) in the successive measurements. I strongly request the authors to provide prospective data if it is available. This makes it more interesting for the readers like me, I believe.

Page 8, table 1: it is better to keep a footnote under table one and explain your abbreviation.

Page 9, after table 2.1, there is table 3..2: Please make it clear

Page 9: In table 3..2, There is β1,  β2,  β3; What are they.

Page 9: I think if you make it clear what is β1, I   and what is β1, II-III. I know it means GMFCS levels, but please mention it under the table.

References:

DOI is missing for some articles like 4, 14, 24 etc.; please check

Author Response

Dear reviewer,

This report summarizes the authors’ replies to each comment or suggestion (marked in bold) of the reviewers by point-by-point answers (marked in blue font). Changes and adaptations in the manuscript are listed in red with the specific lines and indicated with track-changes in the manuscript_track changes.doc

Reviewer 3 Report

The authors are to be commended for this highly significant piece of work which will contribute greatly to the understanding of muscle function over time in growing children with cerebral palsy. The innovative inclusion of breakpoints provides immediate clinical utility by challenging the assumption that the muscle growth trajectory for medial gastrocnemius is both linear and aligned with skeletal growth. This knowledge can help the clinician to clinically reason through the timing and effects of available treatment options to stimulate muscle growth where possible. 

The study design is not clearly stated. Is this observational data which has been organised, rather than an intentional prospective study design? If so, own it, state it up front in the methods section and address this further in the limitations section.

Whilst the data has been collected prospectively and longitudinally, the merging of GMFCS II and III data to maximise the number of observations, raises questions, which again should be discussed more fully in your limitations section. The two comparison groups (GMFCS I and GMFCS II+III) are functionally quite disparate. Children functioning at GMFCS II are independently ambulant, whereas those functioning at GMFCS III use a gait aid with likely lower limb weakness and potentially reduced lower limb weightbearing all of which may affect morphological measures. Additionally, there is a significantly higher proportion of unilaterally involved children in both GMFCS I and GMFCS II levels, compared to GMFCS III. The unbalanced topographical nature within these two groups is important, as you might actually  be comparing higher functioning unilateral involvement with more impaired bilateral involvement (is it possible to include Fig S3 in the published paper rather than reference to supplementary data? - depending on the number of illustration limitations). Future analyses with increased numbers in the bilaterally involved group would be very interesting to clinicians.  

I am less familiar with the statistical methods used in this paper, however this is a strong and reputable research group. If required, the editors may wish to engage a statistician for further review of this section.

The paper is well written, however I have made some annotations for the authors to consider to improve readability in some areas.

Thankyou for the opportunity to review this excellent body of work, which will greatly progress clinical understanding of muscle growth in children with cerebral palsy.
